# Female Brain and Endocrinological Research–Veteran (FemBER-Vet) study: A study protocol for identifying endocrinological, lifestyle and psychosocial determinants of brain health outcomes in female veterans for future intervention success

**Tamlyn Watermeyer**[1,2,3,4]* , **Elliott Atkinson**[1,4], **Glyn Howatson**[1,4,5], **Gill McGill**[1,4], **Christina Dodds**[1,4], **Paul Ansdell**[1,4], **Chinedu Udeh-Momoh**[3,6,7,8,9]

**1** Faculty of Health & Life Sciences, Northumbria University, Newcastle, United Kingdom, **2** Edinburgh Dementia Prevention, University of Edinburgh, Edinburgh, United Kingdom, **3** Female Brain & Endocrine Health Research (FemBER) Consortium, **4** Northern Hub for Military Veterans and Families, Northumbria University, Newcastle, United Kingdom, **5** Water Research Group, North West University, Potchefstroom, South Africa, **6** Ageing Epidemiology (AGE) Research Unit, School of Public Health, Imperial College London, London, United Kingdom, **7** Brain and Mind Institute, Aga Khan University, Nairobi, Kenya, **8** Division of Clinical Geriatrics, Center for Alzheimer Research, Karolinska Institutet and Karolinska University Hospital, Stockholm, Sweden, **9** Division of Public Health Sciences, Wake Forest University School of Medicine, Winston-Salem, North Carolina, United States of America

☯ These authors contributed equally to this work.

\* tamlyn.watermeyer@northumbria.ac.uk

## Abstract

### Background

Recent studies have demonstrated a greater risk of dementia in female veterans compared to civilians; with the highest prevalence noted for former service women with a diagnosis of psychiatric (trauma, alcoholism, depression), and/or a physical health condition (brain injury, insomnia, diabetes). Such findings highlight the need for increased and early screening of medical and psychiatric conditions, and indeed dementia, in the female veteran population. Further, they call for a better understanding of the underlying biopsychosocial mechanisms that might confer heightened risk for female veterans, to tailor preventative and interventional strategies that support brain health across the lifespan.

### Methods

The Female Brain and Endocrinological Research–Veteran (FemBER-Vet) Study will create a highly-phenotyped readiness cohort of ex-service persons as well as non-veterans to assess the impacts of, and risks associated with, military service on brain health, using state-of-the-art non-invasive cognitive, physiological and biomarker capture techniques. FEMBER-Vet will include 90 participants across three study groups (30 female veterans, 30 male veterans, 30 female civilians) to delineate the precise biological, socio-demographic,

**Data Availability Statement:** No datasets were generated or analysed during the current study. All relevant data from this study will be made available upon study completion.

**Funding:** This study is funded by the Office for Veterans' Affairs [G2-SCH-2022-11-12245]. The funders have had no role in the design, implementation of the study nor will they have a role in the future analysis or write up of this work.

**Competing interests:** The authors have declared that no competing interests exist.

health, lifestyle, military-related, and life-course determinants of brain health outcomes (psychosocial, cognitive, neurophysiological, and other biomarkers).

## Discussion

This work addresses the poorly understood biopsychosocial outcomes that female veterans experience compared to their male counterparts and the general female population. Ultimately, it will provide evidence to support the development of tailored interventions for an emerging health priority that currently lacks sufficient evidence for screening and therapeutic intervention.

## Introduction

There is growing evidence that female veterans show distinct, and poorer health outcomes than their male veteran counterparts and matched women from the general population [1, 2]. Brain health is an outcome of emerging importance in an ageing society where frailty and dementia present unprecedented social and financial challenges. Brain health encompasses biology (biomarkers of neural integrity), cognition, emotions and motor function (mobility) and is assumed to be dynamic, improving and deteriorating over time or the lifespan, contingent on the presence or modification of risk and protective factors [3]. In biological women, ageing can further influence brain health outcomes through the cascade of endocrinological events as they transition through pre-, peri- and post-menopausal life stages [4].

Recent studies from the USA demonstrated a greater risk, and earlier onset of, dementia in female veterans compared to male veterans, even when controlling for competing risk of death [1, 2]. Moreover, female veterans with a diagnosis of psychiatric (trauma, alcoholism, depression), and/or physical health condition (brain injury, insomnia, diabetes), show an increased risk for developing dementia than those female veterans without such diagnoses [5, 6]. Such studies highlight the need for increased and earlier screening of medical and psychiatric conditions, and indeed dementia, in the female veteran population. They also call for better understanding of the underlying psychobiological and social mechanisms that might confer heightened risk for this group. Such knowledge which will better prepare preventative and interventional strategies that support brain health in the female veteran population.

In the general population, being biologically female is an established risk factor for Alzheimer's disease (AD), which might reflect disparate hormonal events throughout the lifespan influencing female susceptibility to AD-related neuropathology [7–9]. Apart from endocrinological differences to support sex-specific fertility functions, hormones act on other systems to influence health and behaviour. A notable example is how hormones mediate the response to stress [10–12]. Stress, defined as environmental challenges, both internal (e.g. psychological, physiological) and external (e.g. physical injury), exceeding the natural regulatory capabilities of an organism, is also associated with increased risk for cognitive and psychosocial impairment. Importantly, increased prevalence of stress-related conditions has been reported in women relative to men [13].

In both men and women, stress affects physiological responses through two established pathways: the hypothalamus-pituitary-adrenal (HPA) axis and the sympatho-adreno-medullary axis (SAMA). Differences in HPA axis and SAMA-related responses, as well as the function of the stress neuropeptide corticotrophic-releasing factor (CRF), have been observed between the sexes. Womn, in particular, exhibit heightened sensitivity to increased CRF

exposure during stressful episodes [14]. Additionally, sex-dependent variations in the modulation of the stress response by the hormone glucocorticoids are evident [4, 15, 16]. Dysregulated stress responses have been implicated in the development of various neuropsychiatric conditions, including major depressive disorder, post-traumatic stress disorder, and Alzheimer's disease. Female veterans, through their military service, may encounter acute and chronic psychological stressors [17], alongside physical stressors such as traumatic brain injury [18], which has been proposed as another risk factor for cognitive impairment [19]. The impacts of these stressors may differ between sexes in terms of depression and quality of life, as evidenced within military populations [20].

Despite these findings, scant research has critically examined the unique array of individual, health- and military-related risks facing female veteran brain health beyond deployment, and indeed in later life. Even fewer studies have considered a multi-modal approach that encompass the exploration of biological, psychosocial, and lifestyle determinants to inform biomedical and/or psychosocial interventions. This project will address this gap in the evidence-base and delineate the impacts of, and risks associated with, military service in female UK veterans. We hypothesise that female veterans will have poorer brain health outcomes compared with age- and education- matched women from the general population and male veterans. We will explore which biological, clinical, health, lifestyle and social determinants influence brain health in female veterans compared to their male and female civilian counterparts.

## Materials & methods

The aims of this study are i) assess differences in brain health outcomes, as defined below, in female military veterans, male veterans and female civilians; ii) explore determinants of brain health outcomes in female veterans relative to male veterans and female civilians and iii) create a readiness cohort of female and male veterans for future longitudinal and interventional research.

### Study design, status, setting and timelines

This is a 24-month cross-sectional study hosted at Northumbria University, Newcastle, UK. Initial pilot testing commenced 18 October 2023 and data collection will continue until 23 January 2025. Data and Sample processing will begin in February 2025. Results will be available approximately April 2025, once biomarker analysis can be incorporated with neurophysiological, neuropsychological, and secondary outcome data.

### Sample & screening procedures

Female veterans, male veterans and female non-veteran participants aged 35+ years. Using the effect size from van't Wout et al. (2017) [21] who demonstrated efficacy for a therapeutic tool for treating post-traumatic stress disorder (cohen's d = 0.380) and assuming moderate reliability (r = 0.5), with Alpha = 0.05, Power (1-B) = 0.80, a 2-way ANOVA (female veterans/male veterans/female controls) with measures assessed twice (pre vs. post intervention) would require a minimum sample size of 72 participants (24 per group). Considering a 25% attrition rate, this prescribes a final sample size of 90 participants (30 in each group). This sample permits a baseline cross-sectional comparison between groups and allows testing of targeted therapeutic interventions in future.

Recruitment of military veteran participants will be supported by The Northern Hub for Veterans and Military Families Research at Northumbria University as well as regional military veteran stakeholders (e.g. veteran breakfast clubs; virtual hubs; veteran community services). Recruitment of female control participants will be supported by existing participant registries available through Northumbria University as well as social-media advertisements.

All potential participants will be screened, using the eligibility criteria below before being invited to participate.

## Inclusion criteria

1. History of active military service in the United Kingdom (for veterans only).

2. Thirty-five years or older.

3. Have a history of educational and/or work experience to ensure that congenital learning disabilities with intellectual dysfunction are highly unlikely as determined by the Principal Investigator(s) or designee.

4. Be fluent in and able to read and write in English; be willing and able to give written informed consent: and have adequate hearing and visual acuity to complete the required neuropsychological tests.

5. Be in satisfactory physical health and medically stable based on self-reported medical history to be able to partake in neurophysiology and biospecimen assessments. For any abnormalities identified through neurophysiology and/or biospecimen capture that are outside the normal reference ranges, the participant may be included only if the Principal Investigator(s) judge(s) the abnormalities or deviations from normal to be not clinically significant.

6. Demonstrate willingness to complete neuropsychological, neurophysiological and biospecimen and background assessments.

7. Demonstrate capacity to consent to research.

## Exclusion criteria

1. Any contraindications for taking part in neurophysiology and/or biospecimen assessments, such as: pace-makers and implanted cardiac defibrillators (TMS only); clinically significant infection within 30 days (i.e. requiring medical attention) (Biospecimen only); history of neurological illness (TMS only); metal implants within the skull (TMS only); implanted neurostimulator (TMS only); cochlear implants (TMS only)

2. Commencement of, or significant increased dosage (>10mg) of, neuroactive medication (e.g. SSRI) within the last 4 weeks.

3. Female participants must not be pregnant at time of participation.

4. Major surgery, requiring general anaesthesia within 8 weeks before Research Assessment Visit and/or has not fully recovered from surgery.

5. Exposure within the past 6 months to the neuropsychological measures that will be used in FemBER-Vet Study that could influence participant performance (i.e. learning effects).

6. Unable to comply with the study-specific requirements.

## FemBER-Vet registry

Following completion of the study, all veterans will be invited to join our participant registry to receive invitations to subsequent studies and learn of research opportunities with institutional, research and stakeholder partners. Participants will complete a contact form with their preferred contact mode and contact type (newsletters, email, etc).

### Ethics approval, consent procedures and participant reimbursement

The study received full ethical approval from the Northumbria University Research & Ethics Committee (Ref: 3438). Participants are provided with a Participant Information Sheet (PIS) prior their research visit, where the researcher responsible for taking consent reviews the PIS systematically, allowing time for discussion and queries. Written informed consent is taken in keeping with the Declaration of Helsinki. Following completion of the research visit, participants receive a £30 gift voucher as an appreciation for their time and commitment, as well as reimbursement for travel expenses.

### Data collection and outcome measures

**Background.** The following socio-demographic variables will be collected for all participants regardless of veteran status: age, biological sex at birth, identified gender, sexual orientation, ethnicity, years and level of education; household income, occupational history. The "LGBT ban" in the UK military was officially lifted in 2000, following the implementation of the European Convention on Human Rights, which led to changes in UK legislation to protect the rights of LGBT individuals, including their right to serve in the military without fear of discrimination based on sexual orientation or gender identity. Prior to this, there were policies in place that prohibited lesbian, gay, bisexual, and transgender (LGBT) individuals from serving openly in the armed forces. As we are interested in lifetime and occupational stressors, such as bullying, harassment and discrimination (see below), we will ask participants for their identified gender and sexual orientation, as well as ethnicity, to understand whether these variables influence outcomes relating to psychosocial or other outcomes. Participants are reminded during the interview that they do not have to disclose information regarding these protected characteristics if they do not feel comfortable to do so and such refusal will not affect their participation in the study.

For military veterans, military-related information will be collected: serving status; length of service; service branch/unit; rank; number of deployments and their geographic regions; engagement type (regular, reserve), and combat role, if any.

**General health.** All participants will complete a general health questionnaire that includes self-reported items such as: height, weight, history of past and current medical conditions, including neurodevelopmental conditions (such as Attention Deficit/Hyperactivity Disorder); current and past medications. All participants will be asked about history of Traumatic Brain injury (number of injuries, age at injury/ies); military veterans will be further prompted to report if their injury/ies were related to military service.

**Endocrinological and reproductive health.** Female participants complete questions relating to endocrinological health and fertility, such as menopausal status, menopausal and menstrual symptoms, hormonal contraception and other hormonal use (e.g. hormonal replacement therapy), history of reproductive surgeries (e.g. hysterectomy); age of menarche and menopause (if applicable), history of pregnancies and pregnancy-related health conditions (e.g. gestational diabetes, hyperemesis gravidarum) and sexual health conditions affecting fertility. Male participants will complete questions relating to endocrinological health, such as hormone use, number of pregnancies fathered (both live and non-live births) as well as history of reproductive surgeries and sexual health conditions affecting fertility.

All participants, regardless of veteran status, will complete the following neuropsychological and psychosocial, biological and neurophysiological assessments:

**Neuropsychological and psychosocial assessments.** The following measures will be employed to assess cognitive function.

The Repeatable Battery for Assessment of Neuropsychological Status [RBANS, 22] which includes 12 subtests to assess domains of attention, memory, visuospatial/construction and language ability.

a. The Preclinical Alzheimer Cognitive Composite [PACC, 23] which is a selection of 4 sub-tests weighted towards episodic memory. It comprises four components: the Logical Memory I and II [extracted from the Wechsler Memory Scale-Revised, 24], the Free and Cued Selective Reminding Test [FCSRT, 24, 25], the Wechsler Adult Intelligence Scale-IV (WAIS-IV) Coding subtest [24], and the Mini-mental State Examination [MMSE, 26].

b. The Binding in Neuropsychiatric Disorders Visual Short-Term Memory Binding Test [BIND-VSTMB, 27] is a web-based self-administered task that assesses conjunctive memory binding ability, the ability to integrate different pieces of information (e.g. a shape and its' colour) into a cohesive memory representation [28].

c. The Four Mountains Task [4M, 29] is an allocentric spatial memory test. Memory for the topographical layout of four mountains within a computer-generated landscape (on iPad) is tested using a delayed match-to-sample paradigm.

The following measures will be employed to assess psychosocial parameters:

a. The Center for Epidemiological Studies-Depression Scale [CES-D, 30] will be used to assess depressive symptoms. The CES-D is a 20-item measure that asks participants to rate how often over the past week they experienced symptoms associated with depression, such as restless sleep, poor appetite, and feeling lonely. The State-Trait Anxiety Inventory (STAI) form Y [31] will be used assess anxiety symptoms. State anxiety items include: "I am tense; I am worried" and "I feel calm; I feel secure." Trait anxiety items include: "I worry too much over something that really doesn't matter" and "I am content; I am a steady person."

b. Lifetime trauma, early life stress, post-traumatic stress symptoms will be assessed using the Life Events Checklist for DSM-5 [LEC5, 32] is a self-report questionnaire created to identify potentially traumatic events experienced by individuals throughout their lives. It evaluates exposure to 16 specific events recognized to potentially lead to PTSD or significant distress. Additionally, it includes an extra item to gauge any other exceptionally stressful event not covered by the initial 16 items. In addition, adverse childhood events before the age of 18 will be assessed using six items from Yang, Quach [33]. These items comprise yes/no responses to the following questions: 'Before age 16, would you say your family during that time was pretty well off financially, about average, or poor?'; 'While you were growing up, before age 16, did financial difficulties ever cause you or your family to move to a different place?'; 'Before you were 18-year old, did you have to do a year of school over again?'; 'Before you were 18-year old, were you ever in trouble with the police?'; 'Before you were 18-year old, did either of your parents drink or use drugs so often that it caused problems in the family?' and, 'Before you were 18-year old, were you ever physically abused by either of your parents?'.

c. Alcohol Misuse will be assessed using the Alcohol Use Disorder Identification Test [AUDIT, 34] is a 10-item alcohol harm screening measure developed by the World Health Organisation and adopted by several healthcare settings as well as other veteran studies [35, 36].

The following measures will be employed to assess lifestyle variables:

1. Sleep will be assessed using the Pittsburgh Sleep Quality Index [PSQI, 37] for sleep quality and the 10-item Berlin Sleep Questionnaire [BSQ, 38] for daytime sleepiness. The PSQI is

an 8-item scale comprising subscales of sleep latency, sleep duration, habitual sleep efficiency and subjective quality. Scores range from 0–3 and are summed to yield a total score raging 0–12; higher scores indicative of worse overall sleep quality. The BSQ is a 10-item questionnaire that asks questions about snoring, daytime fatigue and breathing during sleep. Scoring is divided into categories and is contingent on participant responding. Category 1 is positive with 2 or more positive responses to questions 2–6; Category 2 is positive with 2 or more positive responses to questions 7–9; Category 3 is positive with 1 or more positive responses and/or a BMI>30; Final Results: 2 or more positive categories indicates a high likelihood of sleep apnoea.

2. Physical Activity will be assess using the International Physical Activity Questionnaire Short Form [IPAQ-SF, 39]. This includes a question on the duration and frequency spent sitting, walking, partaking in moderate activities and vigorous activities.

3. Current diet will be assessed by asking participants the following question: Compared to a month ago, have you managed to keep a healthy diet, for example eating fresh fruits and vegetables (response options: "Yes, I always have a healthy diet"; "Overall, my diet has improved since a month ago"; "Overall, my diet has been worse since a month ago"; "No, I always have an unhealthy diet").

4. Smoking Status & History will be assessed by asking participants the following questions: Do you currently smoke? ("No, have never smoked"; "No, just have tried once or twice"; "No, stopped smoking"; "Yes, only occasionally"; "Yes, on most or all days"); How old were you when you first started smoking? (Age; "Do not remember"); How old were you when you stopped smoking? (Age; "Do not remember") What did you usually smoke? (Cigarettes; Cigars; Pipe; None of the above; other).

5. Subjective Cognitive Complaints will be assessed by asking participant to response yes/no to the following questions: Have you noticed changes to your memory ability/ thinking ability (e.g. concentration, decision-making)/orientation to time (e.g. you confuse morning and evening)/orientation to space (e.g. you get lost more often in familiar places)?

**Biological assessments.** Participants will be asked to fast from approximately 20:00 the evening before their research visit scheduled between 07:30–09:30. Fasting is to be confirmed verbally with the participant and any deviations noted. Participants will receive breakfast following the biological capture but are not offered caffeinated beverages until completion of neurophysiology.

Subject to funding, time constraints and expertise available at the research site, analyses will be conducted in-house or via external partners (participants are informed of and consent to the possibility of external collaboration for biological data analysis). All biological samples receive separate HTA identification numbers, preserving confidentiality. The use of biologically based markers for AD detection (blood- and saliva-based biomarkers, in particular) is a rapidly evolving research landscape [40]. Therefore, the following methods, procedures and biomarker targets may change to incorporate new technologies, methodological advances and future literature.

a. *Blood sample processing*. A total of 40ml of blood will be drawn primarily from antecubital vein into one 10 ml red top serum tube and three separate 10ml purple top EDTA vacutainers by suitably trained personnel for analyses of blood-based biomarkers including endocrinological and immune markers (e.g. estradiol, progesterone, testosterone levels, Il-6, TNFa), and AD-relevant biomarkers (e.g. Aβ-40, Aβ-42, BDNF, GFAP, NFL, brain derived

tau, total tau, pTau181/217). Samples will be centrifuged at 4˚C (preferred) or room temperature at 1500–3000 x g for 5–15 minutes until plasma and cells or until serum and clots are separated. The samples are then aliquoted in 0.5ml volume per aliquot and then stored at -80˚C until down-stream biomarker analyses are performed. These processes occur immediately or at least within 2 hours of collection.

b. *Saliva*. Saliva ascertainment will involve the participant, via simple passive drool method collecting a morning and evening saliva sample, using the patented SimplOFy™ Saliva Collection Kits [Catalog Number SIMPL-302]. Spools will be used for analysis of AD-related saliva biomarkers (e.g. lactoferrin, amyloid beta and neurofilament-light-chain), cortisol and other steroid hormones and inflammatory markers. Participants complete the self-administered morning capture using the device at the research visit. They are asked to repeat this procedure at-home on the day of their visit and return the sample to the research team via pre-paid postage and packaging service that meet regulatory compliance. Following administration or receipt, devices are stored at −80˚C until analyses are performed. Returned samples are processed and stored identically to research-visit samples.

c. *Hair*. Hair samples will also be collected at the morning time point, to evaluate endocrine and metabolic biomarkers expressed chronically. For this purpose, participants provide a strand of their hair (about 1/2 pencil thick) cut with scissors as close as possible to the scalp. The hair sample will be wrapped in tin foil and stored for future analysis.

**Neurophysiological assessment.** Neurophysiological testing will be performed by a trained administrator and involve the use of single- and paired-pulse transcranial magnetic stimulation, in conjunction with percutaneous electrical stimulation of peripheral nerves. In addition to ruling out the exclusion criteria for TMS, the criteria from Rossi, Antal [41] will be identified before assessments. All stimulations will be delivered with the participant resting, with any trials containing pre-stimulus muscle contraction disregarded.

a. *Electrical nerve stimulation*. The maximum muscle compound action potential (Mmax) will be quantified by delivering 0.2 ms stimuli to the median nerve (DS7AH stimulator; Digitimer, Welwyn Garden City, UK). Electrodes will be placed ~2 cm apart over the nerve with the cathode positioned distal to the anode and ~2 cm proximal to the wrist. Single stimuli will begin at an intensity of 10 mA, and increase in 5 mA increments until the evoked potential amplitude plateaus. The intensity at the plateau will be recorded and one further stimulation will be delivered at 120% of this value to ensure a supramaximal stimulus. The Mmax will be recorded and used to normalise subsequent stimulations. Thereafter, a visual contraction threshold will be recorded as the lowest intensity needed to elicit a contraction of the first dorsal interroseous (FDI) muscle (approximately 12–18 mA).

b. *Transcranial magnetic stimulation*. Single and paired pulse magnetic stimuli of 1 ms duration will be delivered over the contralateral motor cortex to the non-dominant hand (postero-anterior intracranial current flow) with a figure of eight coil powered by a BiStim unit with a Magstim 2002 stimulator (The Magstim Company, Whitland, UK). Optimal stimulation location will be determined as the area of stimulation eliciting the greatest motor evoked potential (MEP) in the first dorsal interroseous (FDI). Resting motor threshold (rMT) will be quanitfied as the lowest stimulation intensity (% maximal stimulator output) eliciting a discernable MEP in $\geq$3 out of 5 stimulations.

c. *Surface electromyography*. Evoked potentials will be recorded via a bipolar configuration of Ag-AGCl electrodes placed over the FDI, with a reference electrode placed on the wrist.

EMG will be amplified (×1000, 1902, Cambridge Electronic Design, Cambridge, UK), sampled at 5000 Hz, band pass filtered (20–2000 Hz, Power1401, Cambridge Electronic Design, Cambridge, UK).

The above methodologies will be combined to assess the following neurophysiological variables:

a. Corticospinal tract excitability: will be assessed with single-pulse TMS. The average motor evoked potential amplitude from 20 stimuli delivered at 120% rMT will be quantified and expressed in both mV as well as %Mmax to facilitate between-subjects comparisons.

b. Short-interval intracortical inhibition (SICI): γ-aminobutyric acid (GABA)-ergic inhibition within the motor cortex will be assessed with paired-pulse TMS. First, a subthreshold (80% rMT) pulse will be delivered 2.5 ms prior to a test pulse [120% rMT, 42]. The amplitude of the conditioned MEPs will be expressed as a % of unconditioned MEPs. Twenty conditioned MEPs will be compared with 20 unconditioned MEPs.

c. Intracortical facilitation (ICF): Facilitatory glutamatergic neurotransmission within the motor cortex will be assessed with paired-pulse TMS. First, a subthreshold (80% rMT) pulse will be delivered 12 ms prior to a test pulse [120% rMT, 42]. The amplitude of the conditioned MEPs will be expressed as a % of unconditioned MEPs. Twenty conditioned MEPs will be compared with 20 unconditioned MEPs.

d. Short- and long-latency afferent inhibition (SAI, LAI): The capacity to integrate antidromic sensory impulses with orthodromic motor impulses will be assessed with a combination of electrical nerve stimulation and TMS. First, an electrical stimulus will be delivered at 120% of the intensity that produces a visual contraction of the FDI [43, 44]. Thereafter, a single TMS pulse will be delivered 20ms after the electrical stimulation at 120% rMT for SAI, whereas the interstimulus interval will be 200 ms for LAI [45]. The amplitude of the conditioned MEPs will be expressed as a % of unconditioned MEPs. Twenty conditioned MEPs will be compared with 20 unconditioned MEPs.

In total, participants will receive 100 transcranial magnetic stimuli throughout the neurophysiological testing, this number is considered low risk by recent guidelines [41].

**Military-specific psychosocial measures.** This study will pilot the use of items from the Deployment Risk and Resilience Inventory-2 questionnaires [DRRQ-2, 46] in military participants only. With support from veteran steering members and feedback from pilot veteran participants, the language of these items has been adapted to suit the British military context and the questioning timeframe will be adapted to include overall military service, rather than specific deployments. The purpose of including these items in the current study is to explore themes related to military service that may serve as the basis for themes and research questions in future qualitative studies. We will use and adapt the following questionnaires: Deployment environment; Combat Experiences; Exposure to Nuclear, Biological, or Chemical Agents; Post-battle Experiences; Support from Family/Friends and Relationships During Deployment. We will solicit feedback from participants regarding our adaptions and the questionnaire themes which include bullying, sexual and physical harassment as well as environmental and psychological exposures.

### Statistical analyses plan

Descriptive statistics for the overall sample as well as the separate participant groups (female veterans, male veterans, female civilians) will be reported for all outcome measures. For the

main hypothesis testing, normal distribution of data will be assessed with the Shapiro-Wilk test, and if normal distribution is confirmed, independent samples t-tests or one-way ANOVAs will be conducted to compare participant groups on demographic, neuropsychological, biomarker and neurophysiological outcomes. If the variables do not follow a normal distribution, the subsequent statistical tests will be employed: for dichotomous variables, a chi-square test will be utilized to ascertain differences between groups, whereas, for continuous variables, a Kruskal-Wallis's test will be applied for between-group comparisons. A p-value of $<0.05$ will be deemed significant. Female veterans and age- and education-matched female civilians will be compared on female-specific endocrinological, epi/genetic markers using chi-square and t-tests. For exploratory hypothesis testing, we will use multivariate regression analyses to determine which demographic, biological, clinical, health, lifestyle and social determinants influence neuropsychological, biomarker and neurophysiological outcomes. Exploratory moderation and/or mediation analyses will be included to assess potential bidirectional associations between, endocrinological, epi/genetic, AD-related biomarkers and inflammation (e.g. IL-6, TNFα) markers, and their associations with neuropsychological, psychosocial, and neurophysiological outcomes. We will explore these relationships within the overall sample and within participant groups. Bonferroni corrections will be applied where required to account for type 1 error. Effect sizes will be reported to support development of larger longitudinal work. Optional qualitative feedback obtained from participants regarding the use and themes of the DRRI-2 will be summarised, but primarily used for the development of future qualitative projects.

## Data management and sharing

To ensure participant privacy and confidentiality, personal data is stored securely in password-protected files, safeguarded against unauthorized access by external parties. Access to raw data and materials is restricted to project-team members only. Each participant is assigned a randomly generated code that does not reveal their identity, and this code is exclusively used for file and sample naming. Following a brief embargo period, all relevant data from this study will be made available upon study completion through the Alzheimer's Disease Data Initiative platform. Researchers may request access through application to our data management committee.

## Discussion

Brain health research in female military populations is virtually non-existent, thereby precluding effective evidence-based treatment or support practices for this underserved population. Reasons for the reported discrepancies in dementia prevalence and incidence between female veterans with male veterans and general female populations [1, 2, 5, 6] might be attributed to unique psychological and physical stressors associated with military service leading to poorer health outcomes. Furthermore, other contributors might be sex-specific stressors as well as biological differences, including different endocrinological responses, to such stressors between the sexes [4]. Therefore, the novelty of the project lies in its methodological design and ambition. To date, the few female veteran studies have primarily focussed on psychosocial health [e.g. 47–51]. We aim to delineate the precise biological, socio-demographic, health, lifestyle, military, and lifetime determinants of broader brain health outcomes using innovative and non-invasive cognitive, physiological and biomarker capture techniques. This will allow our research team and others to design future intervention trials addressing all aspects of brain health. Such interventions could be psychosocial, lifestyle-based, cognitive or a combination thereof, which will invite future collaborations with Armed Forces stakeholders and the NHS.

Currently, NHS provision for veterans is focussed on prosthetics and mental health support services [52]. For the latter, mainstream, complex and high-intensity trauma-focussed therapies are offered but in practice, these services do not operate under a wider strategy to prevent or address compromised brain health, of which mental health is only one component. For example, efforts to prevent the emergence or mitigate the impact of AD dementia in veterans could reduce further burden on NHS services and benefit veteran communities. This approach also takes into account the underpinning principles of The Armed Forces Covenant, signed by NHS Trusts as part of a pledge to priorities any service-related health issue. Our project is a first step in a programme of research in the UK that will help inform such strategies to support female veterans. Furthermore, it will inform preventative strategies for women who serve in the future, for the benefit of veteran community overall.

## Conclusion

To our knowledge, this is the first study to explore the precise biopsychosocial mechanisms of brain health in the context of female military veterans. Long-term, the FemBER-Vet project aims to develop a highly phenotyped readiness cohort to provide novel psychobiological data and assess the efficacy of future interventions tailored to this underserved population.

## Acknowledgments

The authors wish to thank NUTRAN at Northumbria University for access to clinical test rooms. The study team also wishes to thank the Barbour Foundation for their generous donation towards our participants' travel and accommodation reimbursements.

## Author Contributions

**Conceptualization:** Tamlyn Watermeyer, Paul Ansdell, Chinedu Udeh-Momoh.

**Data curation:** Tamlyn Watermeyer, Elliott Atkinson, Paul Ansdell, Chinedu Udeh-Momoh.

**Funding acquisition:** Tamlyn Watermeyer, Glyn Howatson, Gill McGill, Christina Dodds, Paul Ansdell, Chinedu Udeh-Momoh.

**Investigation:** Tamlyn Watermeyer, Elliott Atkinson, Paul Ansdell, Chinedu Udeh-Momoh.

**Methodology:** Tamlyn Watermeyer, Paul Ansdell, Chinedu Udeh-Momoh.

**Project administration:** Tamlyn Watermeyer, Elliott Atkinson.

**Resources:** Tamlyn Watermeyer, Paul Ansdell, Chinedu Udeh-Momoh.

**Supervision:** Tamlyn Watermeyer, Paul Ansdell, Chinedu Udeh-Momoh.

**Writing – original draft:** Tamlyn Watermeyer.

**Writing – review & editing:** Tamlyn Watermeyer, Elliott Atkinson, Glyn Howatson, Gill McGill, Christina Dodds, Paul Ansdell, Chinedu Udeh-Momoh.

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
