## [Editor Report · Decision Letter 0]

28 Jun 2024

PONE-D-24-18322Female Brain and Endocrinological Research – Veteran (FemBER-Vet) Study: A study protocol for identifying endocrinological, lifestyle and psychosocial determinants of brain health outcomes in female veterans for future intervention successPLOS ONE

Dear Dr. Watermeyer,

Thank you for submitting your manuscript to PLOS ONE. After careful consideration, we feel that it has merit but does not fully meet PLOS ONE’s publication criteria as it currently stands. Therefore, we invite you to submit a revised version of the manuscript that addresses the points raised during the review process. Thank you for providing us with all the documents we have requested, including peer-reviewed external funding.

Because your protocol has already undergone peer review as part of the funding process, I do not think we need to send your manuscript out for additional peer review. However, there is one minor issue that needs to be addressed before we can accept your protocol for publication:

Please could you change "female” or "male" to "woman” or "man" as appropriate, when used as a noun (see for instance https://apastyle.apa.org/style-grammar-guidelines/bias-free-language/gender).

We look forward to receiving your revised manuscript.

Kind regards,

Steve Zimmerman, PhD

Senior Editor, PLOS ONE

Journal Requirements:

2. Thank you for stating the following financial disclosure: "Office for Veteran Affairs [G2-SCH-2022-11-12245]."

---

## [Editor Report · Decision Letter 1]

26 Jul 2024

PONE-D-24-18322R1Female Brain and Endocrinological Research – Veteran (FemBER-Vet) Study: A study protocol for identifying endocrinological, lifestyle and psychosocial determinants of brain health outcomes in female veterans for future intervention successPLOS ONE

Dear Dr. Watermeyer,

Thank you for submitting your manuscript to PLOS ONE. After careful consideration, we feel that it has merit but does not fully meet PLOS ONE’s publication criteria as it currently stands. Therefore, we invite you to submit a revised version of the manuscript that addresses the points raised during the review process.

Although you indicated that you have revised the manuscript, there are some issues with the files that form your submission:

1) The file called "manuscript" is the original manuscript, not the revised ms. Please replace this with an unmarked version of your revised paper without tracked changes. You should upload this as a separate file labeled 'Manuscript'.

2) The file called "Revised Manuscript with Track Changes" does appear to be a revised manuscript, but (a) there are no track changes, and (b) "female" is still used as a noun in several places. Please replace this with a marked-up copy of your manuscript that highlights changes made to the original version. You should upload this as a separate file labeled 'Revised Manuscript with Track Changes'.

We look forward to receiving your revised manuscript.

Kind regards,

Steve Zimmerman, PhD

Senior Editor, PLOS ONE
---

## [Author Response · Author response to Decision Letter 1]

26 Jul 2024

Dear Dr Zimmerman,

RE: PLOS ONE Decision: Revision required [PONE-D-24-18322] - [EMID:b6c6dd189fed26f4]

Thank you for your further comments, please see responses below, in turn. 

1) The file called "manuscript" is the original manuscript, not the revised ms. Please replace this with an unmarked version of your revised paper without tracked changes. You should upload this as a separate file labeled 'Manuscript'.

Apologies, I did not realise I also need to provide a clean version again at this stage, I have updated this as a clean version now. 

2) The file called "Revised Manuscript with Track Changes" does appear to be a revised manuscript, but (a) there are no track changes, and (b) "female" is still used as a noun in several places. Please replace this with a marked-up copy of your manuscript that highlights changes made to the original version. You should upload this as a separate file labeled 'Revised Manuscript with Track Changes'.

I am not sure why you cannot see track changes, as these are showing for me. I have now highlighted these in yellow so they are more obvious.

As with regards to language, I have updated the manuscript in accordance with your preference as much as possible. I have also reviewed the new updated manuscript and run it through AI to identify places where updating of nouns to woman/women are still outstanding. I believe this is now complete as much as possible. However, if you are referring to references such as “ female military veterans” in stead of “women military veterans” where it is being used as a gerund, I am afraid that I am reluctant to update this to the latter. We work closely with our military and LGBTQIA+ stakeholders surrounding language for this project. The draft was subject to review for language on gender by LBTQIA+ pilot military and civilian participants and the term female and male as nouns were chosen to designate sex assigned at birth reference by them. Nonetheless, I have made these changes as you have asked. This preference was shown also by non-pilot participants but who were two members of the trans community. Respectfully, we write for, and are dedicated to upholding the preferences of, our stakeholders and I would be reluctant to make further changes to the gerunds as described above. It is difficult to anticipate all preferences, but I would suggest that the APA guidance is not relevant for local North-East of UK preferences at least amongst our LGBTQIA+ and military stakeholders. 

Sincerely, 

Tam 

Dr Tamlyn Watermeyer

---

## [Editor Report · Decision Letter 2]

5 Aug 2024

Female Brain and Endocrinological Research – Veteran (FemBER-Vet) Study: A study protocol for identifying endocrinological, lifestyle and psychosocial determinants of brain health outcomes in female veterans for future intervention success

PONE-D-24-18322R2

Dear Dr. Watermeyer,

We’re pleased to inform you that your manuscript has been judged scientifically suitable for publication and will be formally accepted for publication once it meets all outstanding technical requirements.

Kind regards,

Steve Zimmerman, PhD

Senior Editor, PLOS ONE
---

## [Editor Report · Acceptance letter]

19 Aug 2024

PONE-D-24-18322R2 

PLOS ONE

Dear Dr. Watermeyer, 

I'm pleased to inform you that your manuscript has been deemed suitable for publication in PLOS ONE. Congratulations! Your manuscript is now being handed over to our production team.

Kind regards, 

on behalf of

Dr Steve Zimmerman 

Staff Editor

PLOS ONE